# Periodicity Decoupling Framework for Long-term Series Forecasting

**Tao Dai**[1,2,4]**, Beiliang Wu**[1]**, Peiyuan Liu**[1,3,*]**Naiqi Li**[3,*]**Jigang Bao**[3]**, Yong Jiang**[3]**, Shu-Tao Xia**[3]

[1]College of Computer Science and Software Engineering, Shenzhen University, China
[2]National Engineering Laboratory for Big Data System Computing Technology, Shenzhen University, China
[3]Tsinghua Shenzhen International Graduate School, Tsinghua University, Shenzhen, China
[4]WeBank Institute of Financial Technology, Shenzhen University, China
{daitao.edu, peiyuanliu.edu, linaiqi.thu}@gmail.com; {jiangy, xiast}@sz.tsinghua.edu.cn

## Abstract

Convolutional neural network (CNN)-based and Transformer-based methods have recently made significant strides in time series forecasting, which excel at modeling local temporal variations or capturing long-term dependencies. However, real-world time series usually contain intricate temporal patterns, thus making it challenging for existing methods that mainly focus on temporal variations modeling from the 1D time series directly. Based on the intrinsic periodicity of time series, we propose a novel Periodicity Decoupling Framework (PDF) to capture 2D temporal variations of decoupled series for long-term series forecasting. Our PDF mainly consists of three components: multi-periodic decoupling block (MDB), dual variations modeling block (DVMB), and variations aggregation block (VAB). Unlike the previous methods that model 1D temporal variations, our PDF mainly models 2D temporal variations, decoupled from 1D time series by MDB. After that, DVMB attempts to further capture short-term and long-term variations, followed by VAB to make final predictions. Extensive experimental results across seven real-world long-term time series datasets demonstrate the superiority of our method over other state-of-the-art methods, in terms of both forecasting performance and computational efficiency. Code is available at https://github.com/Hank0626/PDF.

## 1 Introduction

Time series forecasting plays an essential role in multiple applications, including weather prediction (Angryk et al., 2020), energy management (Zhou et al., 2021), financial investment (Patton, 2013), and traffic flow estimation (Chen et al., 2001). Recently, with the rapid development of deep learning, plenty of deep learning (DL)-based methods have been developed for time series forecasting (Lim & Zohren, 2021), which can be roughly divided into CNN-based (Wang et al., 2022; Liu et al., 2022a) and Transformer-based methods (Li et al., 2019; Zhou et al., 2021).

Existing DL-based methods mainly focus on 1D temporal variation modeling directly, which plays a crucial role in time series forecasting. Among them, CNN-based methods (Bai et al., 2018; Wang et al., 2022; Wu et al., 2023) have shown the powerful ability to capture short-term variations. For example, TCN (Bai et al., 2018) incorporates the local information of time series along the temporal dimensions by utilizing convolution operations, and exhibits superior performance in short-term and medium-term predictions. However, this type of method usually fails to work well for long-term time series, due to the limited representation of long-term dependencies. By contrast, Transformer-based methods (Li et al., 2019; Zhou et al., 2021; Wu et al., 2021) excel at capturing long-term dependencies due to the use of self-attention mechanism. For example, Autoformer (Wu et al., 2021) attempts to exploit the series-wise temporal dependencies with auto-correlation mechanism. PatchTST (Nie et al., 2023) proposes a novel patching strategy to retain local semantic information within each patch. Although the Transformer-based methods have shown more competitive perfor-

---

*Correspondence to: Peiyuan Liu and Naiqi Li

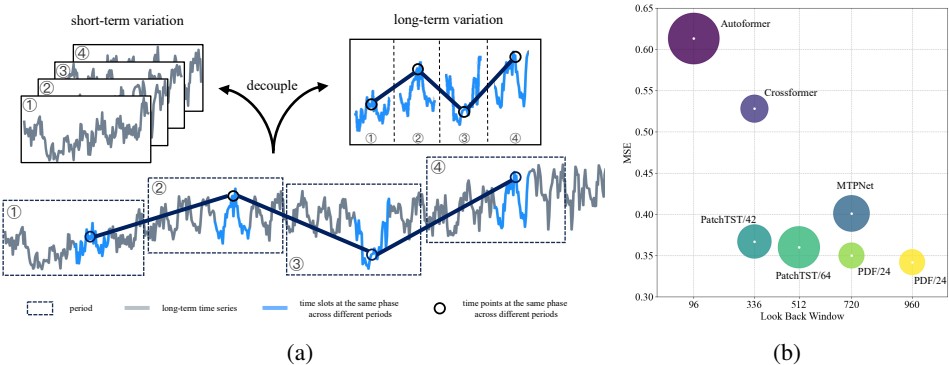

Figure 1: (a) Illustration of periodicity-based decoupling into short- and long-term series. (b) Comparison of MSE and patch number of our PDF over other Transformer-based methods to predict future 96 time steps on Traffic dataset. Tranformer-based methods obtain worse MSE results with more patch numbers. By contrast, our PDF achieves the lowest MSE with only 24 patches on the look-back window of 960 length. The radius of the circle is the number of patches.

mance than CNN-based methods, they often suffer from heavy computational costs, especially for long-term time series input, which thus limits their real applications.

It is worth considering that the modeling of 1D temporal variations can be a complex task due to the intricate patterns involved. These variations can come in various types, including short-term fluctuations, falling, and rising, which can often overlap with each other (see Figure 1a). Despite the challenges, researchers have made significant progress in this area, and the most effective way to model temporal variation remains an open question. However, it is important to note that real-world time series often exhibit multi-periodicity, such as daily and weekly variations for traffic forecasting, which has been confirmed in recent work (Wu et al., 2023). Furthermore, long-term time series can be simplified or decoupled based on a predetermined period. For example, as shown in Figure 1a, the original time series can be decoupled into short-term series and long-term series, which contain short-term changes and long-term dependencies, respectively. By taking these observations into account, we can utilize period information to decouple long-term time series.

Motivated by the above observations, we propose a novel Periodicity Decoupling Framework (PDF) for long-term series forecasting by capturing the intricate periodic information inside the time series. Based on the periodicity of the time series, the original 1D time series can be further decoupled into simpler short and long-term series, which respectively represent the local changes and global correlations of the 1D time series. Due to the diversity of short-term variations (e.g., fluctuation, rising, and falling), we employ "frequency slicing", corresponding to different periods, to divide the look-back window into several sub-sequences. For long-term variations, we utilize "period patching" to extract changes within corresponding time segments across all periods (see Figure 1a). The "period patching" ensures each patch contains rich long-term semantic information.

Technically, we propose a novel **P**eriodicity **D**ecoupling **F**ramework (**PDF**) for long-term time series forecasting. As illustrated in Fig. 2, our **PDF** contains three main components: multi-periodic decoupling block (MDB), dual variations modeling block (DVMB), and variations aggregation block (VAB). Unlike the previous methods that focus on 1D temporal variations modeling, our PDF models 2D temporal variations. Specifically, the multi-periodic decoupling block first decouples the 1D time series into different short- and long-term 1D series based on the period of input series in the frequency domain, followed by further reshaping into 2D tensors with rich short- and long-term variations. After that, the dual variations modeling block attempts to capture short-term and long-term variations from the decoupled 2D tensors, followed by a variations aggregation block to make final predictions. Extension experiments on our PDF confirm its state-of-the-art performance across various long-term time series datasets, in terms of both forecasting performance and computational efficiency. Notably, as seen in Figure 1b, our PDF handles the long-term series (with a look-back window length of 960) better while not sacrificing computational cost (with only 24 patches) than other Tranformer-based methods.

Our main contributions are summarized as follows:

- We propose a novel Periodicity Decoupling Framework (PDF) for long-term series forecasting, which fully captures 2D temporal short-term and long-term variations from the decoupled series in a parallel architecture.

- We propose multi-periodic decoupling block to capture various periods of the input series in the frequency domain. Based on the periodicity of the time series, the 1D time series can be decoupled into simpler short- and long-term series formulated with 2D tensors. To fully capture the short- and long-term variations, we propose dual variations modeling block (DVMB) with short- and long-term variations extractor, which is able to preserve the high-frequency information of short-term changes while exploiting long-term dependencies.

- Extensive experiments demonstrate the effectiveness of our PDF over other state-of-the-art methods across various long-term time series datasets, in terms of both forecasting performance and computational efficiency.

## 2 RELATED WORK

Traditional time series forecasting methods such as ARIMA (Anderson & Kendall, 1976) and Holt-Winter (Hyndman & Athanasopoulos, 2018) offer robust theoretical frameworks but suffer from limitations in handling data with intricate temporal dynamics. Recent years have witnessed milestone achievements of deep learning-based approaches in time series forecasting, which mainly include CNN-based (Wu et al., 2023), and Transformer-based methods (Lim & Zohren, 2021).

Convolutional neural network (CNN) has gained widespread popularity due to its ability to capture localized features (Xia et al., 2017; Zhang et al., 2021; Woo et al., 2023). Many CNN-based time series forecasting methods employ Temporal Convolutional Networks (TCN) to extract local temporal dynamics (Bai et al., 2018; Liu et al., 2022a; Wang et al., 2022), where MICN (Wang et al., 2022) and TimesNet (Wu et al., 2023) are related to our method. Typically, MICN attempts to combine local features and global correlations to capture the overall view of time series with convolution kernels. TimesNet focuses on modeling 2D temporal variations in 2D spaces from the extraction of "intra-period" and "inter-period" variations. However, these methods rely heavily on convolution kernels to model series variations, resulting in limited representations of long-term dependencies. Instead, our method can capture both short- and long-term variations simultaneously with dual variations modeling block.

Another type of Transformer-based method has shown more competitive performance in long-term time series forecasting. With the self-attention mechanism, Transformer and its variant are capable of capturing long-term dependencies and extracting global information (Dosovitskiy et al., 2021; Fan et al., 2021; Ryoo et al., 2021; Liu et al., 2022b). However, their scalability and efficiency are constrained by the quadratic complexity of the attention mechanism. To mitigate this, various techniques are proposed to reduce the complexity of the Transformer. For example, LogTrans (Li et al., 2019) utilizes convolution self-attention to reduce the space complexity. Informer (Zhou et al., 2021) applies distilling strategies to exploit the most crucial keys. Pyraformer (Liu et al., 2021) proposes a pyramid attention design with inter-scale and intra-scale connections. More recent work PatchTST (Nie et al., 2023) employs patch-based strategies to enhance the locality while improving long-term forecasting accuracy. However, existing Transformer-based methods still focus on 1D temporal variation modeling and suffer from heavy computational burden for long-term time series. Instead, we propose a more efficient Periodicity Decoupling Framework (PDF) for long-term series forecasting by fully capturing 2D temporal short-term and long-term variations in a parallel architecture.

## 3 PERIODICITY DECOUPLING FRAMEWORK

### 3.1 THE OVERALL ARCHITECTURE

In time series forecasting, given a historical input series $\mathbf{X}_I = [\mathbf{x}_1, \mathbf{x}_2, \ldots, \mathbf{x}_t]^T \in \mathbb{R}^{t \times d}$, it aims to predict future output series $\mathbf{X}_O = [\mathbf{x}_{t+1}, \mathbf{x}_{t+2}, \ldots, \mathbf{x}_{t+T}]^T \in \mathbb{R}^{T \times d}$, where $t$, $T$ is the number of time steps in the past and future, respectively, where $d > 1$ is the number of dimensions. The overall architecture of our method is shown in Figure 2. In our PDF, due to the complex temporal

patterns, it is the first step to decouple the 1D time series for better variation modeling. To this end, we design a **Multi-periodic Decoupling Block** to learn the periods of input series in the frequency domain and convert the 1D time series into short- and long-term series, followed by reshaping into 2D tensors. Then, the obtained short-term and long-term 2D tensors are fed into serveral **Dual Variations Modeling Blocks** (DVMB) to model short- and long-term variations in a parallel way. Finally, we use a **Variations Aggregation Block** to merge the outputs from all DVMBs to yield the final prediction $\mathbf{X}_O$. More details about our PDF are shown in the following sections.

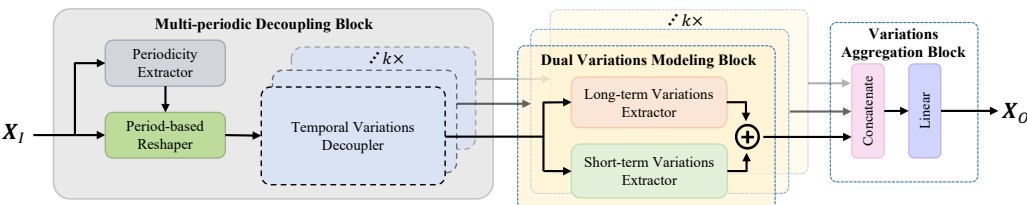

Figure 2: The architecture of our Periodicity Decoupling Framework, which mainly consists of multi-periodic decoupling block, dual variations modeling block, and variations aggregation block.

## 3.2 MULTI-PERIODIC DECOUPLING BLOCK

The Multi-periodic Decoupling block uses Periodicity Extractor and Period-based Reshaper to transform 1D time series into 2D spaces. Then it utilizes Temporal Variations Decoupler to decouple the long-term and short-term information through "period patching" and "frequency slicing".

**Periodicity Extractor.** Previous work (Wu et al., 2023) emphasizes that the original 1D structure of time series inadequately represents only adjacent time point variations, and a 2D structure can effectively capture variations both within and between periods. Therefore, for a given 1D input $\mathbf{X}_I \in \mathbb{R}^{t \times d}$ of dimension d, we employ the Fast Fourier Transform (FFT) (Chatfield, 1981) to analyze the time series in the frequency domain as follows:

$$\mathbf{A} = \text{Avg}(\text{Amp}(\text{FFT}(\mathbf{X}_I))) \tag{1}$$

Here, FFT and Amp denote the FFT and amplitude extraction, respectively. The channel-wise average operation Avg over $d$ channels yields $\mathbf{A} \in \mathbb{R}^t$, representing the amplitudes of $t$ frequencies. Specifically, the $j$-th value $\mathbf{A}_j$ represents the intensity of the periodic basis function for frequency $f$. We use the univariate $X_I \in \mathbb{R}^t$ instead of $\mathbf{X}_I$ to denote the input time series in the following calculation, because the subsequent transformations and predictions are made in a channel-independent manner (Zheng et al., 2014; Nie et al., 2023).

Different from Wu et al. (2023), we select frequencies not only focus on high amplitude but also incorporate those with significant values and amplitude. We assert that frequencies with high amplitude better represent the primary components, while those with larger values facilitate a more discernible distinction between long-term and short-term relationships. We summarize the $k$ frequencies selection by:

$$\mathbf{F}_u = \underset{f_* \in \{1, \cdots, [\frac{t}{2}]\}}{\arg \text{top-}m} (\mathbf{A}), \ \mathbf{F}_{k_1} = \underset{f_* \in \{1, \cdots, [\frac{t}{2}]\}}{\arg \text{top-}k_1} (\mathbf{A}), \ \{f_1, \cdots, f_k\} = \mathbf{F}_{k_1} \cup \text{top-}k_2(\mathbf{F}_u \setminus \mathbf{F}_{k_1}) \tag{2}$$

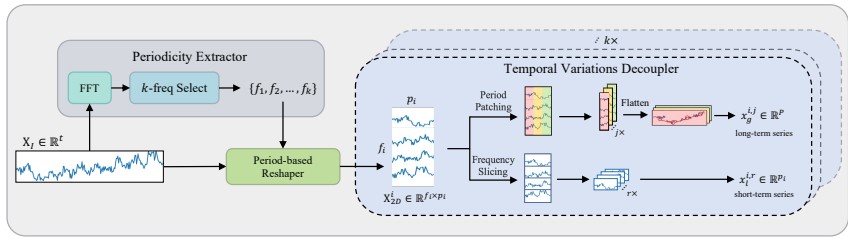

Figure 3: Multi-periodic Decoupling Block.

where $\mathbf{F}_u$ and $\mathbf{F}_{k_1}$ represents the $u$ and $k_1$ frequencies with highest amplitudes from $\mathbf{A}$, respectively. We ensure that $u$ is greater than or equal to $k_1$. Due to the conjugate symmetry in the frequency domain, $f_*$ only focuses on the former $[\frac{t}{2}]$ frequencies. The final set of $k$ frequencies is composed of $\mathbf{F}_{k_1}$ and the top-$k_2$ frequencies with the greatest values from $\mathbf{F}_u \setminus \mathbf{F}_{k_1}$.

**Period-based Reshaper.** Based on the selected frequencies $\{f_1, \cdots, f_k\}$ and corresponding period lengths $\{p_1, \cdots, p_k\}$ ($p_i = \lceil \frac{t}{f_i} \rceil$), we reshape the 1D input series $X_I \in \mathbb{R}^t$ into $k$ 2D tensors by:

$$X_{2D}^i = \text{Reshape}_{f_i, p_i}(\text{Padding}(X_I)), \quad i \in \{1, \cdots, k\} \tag{3}$$

Here, $\text{Padding}(\cdot)$ is employed to extend the length of $X_I$ to $p_i \times f_i$ by filling zeros for $\text{Reshape}_{f_i, p_i}(\cdot)$, where $f_i$ and $p_i$ denote the number of rows and columns of the 2D tensor, respectively. For the obtained 2D tensor $X_{2D}^i \in \mathbb{R}^{f_i \times p_i}$, each row represents the short-term variations and each column represents long-term variations. We then employ **Temporal Variations Decoupler** to decouple the long-term and short-term information through "period patching" and "frequency slicing".

**Period Patching:** Denote the patch length as $p$ and the stride length as $s$, we divide $X_{2D}^i \in \mathbb{R}^{f_i \times p_i}$ along dimension $p_i$ and aggregate along dimension $f_i$ to form a patch. Specifically, $X_{2D}^i$ is patched into multiple patches $x_g^{i,j} \in \mathbb{R}^{N \times P}$, where $N = \lfloor \frac{(p_i - p)}{s} \rfloor + 1$ is the number of patches and each patch contains $P = f_i \times p$ time steps. $x_g^{i,j}$ denotes the $j$-th patch. This patching strategy condenses complete long-term variations between all periods.

Compared with former patching strategies (Nie et al., 2023; Zhang & Yan, 2023), our patches capture a broader scope and richer semantic information, enhancing the capacity of the Transformer for modeling long-term variations. Meanwhile, because the number of patches decreases from $t/s$ to $\max(p_i)/s$, the computational cost is significantly reduced.

**Frequency Slicing:** Along with $f_i$ dimensions, we split $X_{2D}^i$ into several 1D short-term slices $x_l^{i,r} \in \mathbb{R}^{p_i}$, where $r \in [1, f_i]$ denotes the $r$-th row of $X_{2D}^i$. Each local slice represents the short-term variations within every period.

### 3.3 DUAL VARIATIONS MODELING BLOCK

As illustrated in Figure 4, the Dual Variations Modeling Block is composed of two parts: long-term variations extractor and short-term variations extractor. It adopts a dual-branch parallel architecture to model the long-term and short-term variations in the time series. This parallel structure not only better preserves the high-frequency information of short-term changes but also enhances computational efficiency (Wang et al., 2022; Si et al., 2022). The details of each component will be given as follows.

**Long-term Variations Extractor:** Given the patches $x_g^{i,j} \in \mathbb{R}^{N \times P}$ with long-term information, we initially project them into the latent space via a linear projection: $x_g^{i,j} = \text{Linear}(x_g^{i,j}) \in \mathbb{R}^{N \times D}$, where $D$ is the dimension of latent space. Subsequently, $x_g^{i,j}$ will go through several Transformer encoder layers. The specific process of each layer can be described as follows:

$$\begin{aligned} \hat{x}_g^{i,j} &= \text{BatchNorm}(x_g^{i,j} + \text{MSA}(x_g^{i,j}, x_g^{i,j}, x_g^{i,j})) \\ \hat{x}_g^{i,j} &= \text{BatchNorm}(\hat{x}_g^{i,j} + \text{MLP}(\hat{x}_g^{i,j})) \end{aligned} \tag{4}$$

Here, $\text{BatchNorm}(\cdot)$ denotes batch normalization (Ioffe & Szegedy, 2015). $\text{MLP}(\cdot)$ is a multi-layered linear feedforward neural network. Multi-head self-attention $\text{MSA}(\cdot)$ mechanism enhances the representation capacity by employing multiple independent self-attention heads. Each head captures different types of long-term dependencies among different patches. All these heads are combined to obtain more comprehensive dependencies by:

$$X_g^i = \text{Linear}(\text{Flatten}(\hat{x}_g^{i,j})) \in \mathbb{R}^t \tag{5}$$

**Short-term Variations Extractor:** This module contains a sequence of convolution blocks, each consisting of a $\text{Conv1d}$ layer and a non-linear activation function. These blocks are sequentially structured to gradually expand the receptive field, accommodating periods of various lengths. For each local slice $x_l^{i,r}$, the process of each block is:

$$\hat{x}_l^{i,r} = \text{SELU}(\text{Conv1d}(x_l^{i,r})) \tag{6}$$

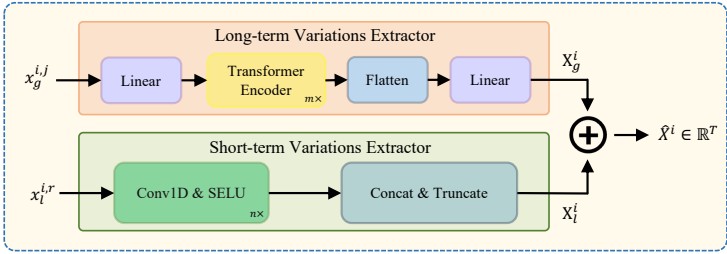

Figure 4: Dual Variations Modeling Block.

where SELU denotes scaled exponential linear units (Klambauer et al., 2017). To get the final prediction of the convolution part, we use the concatenate and truncate operations:

$$X_l^i = \text{Truncate}(\text{Concat}(\hat{x}_l^{i,r})) \tag{7}$$

The final output of the Dual Variations Modeling Block is the summation of $X_g^i$ and $X_l^i$:

$$\hat{X}^i = X_g^i + X_l^i \tag{8}$$

### 3.4 VARIATIONS AGGREGATION BLOCK

The Variations Aggregation Block consolidates the results from $k$ DVMBs. Specifically, we concatenate these $k$ results and then map them through a parameter-shared linear layer to produce univariate prediction $X_O \in \mathbb{R}^T$:

$$X_O = \text{Linear}(\text{Concat}(\hat{X}^i)) \tag{9}$$

The final multivariate prediction $\mathbf{X}_O \in \mathbb{R}^{T \times d}$ is obtained by stacking $d$ univariate prediction $X_O$.

## 4 EXPERIMENTS

**Datasets** We conduct extensive experiments on seven popular real-world datasets (Zhou et al., 2021), including Electricity Transformer Temperature (ETT) with its four sub-datasets (ETTh1, ETTh2, ETTm1, ETTm2), Weather, Electricity, and Traffic. We adopt the same train/val/test splits ratio 0.6:0.2:0.2 as Zhou et al. (2021) for the ETT datasets and split the remaining three by the ratio of 0.7:0.1:0.2 following Wu et al. (2021).

**Baselines** We select representative state-of-the-art methods from the recent LTSF landscape as baselines, including the following categories: 1) Transformer-based models: PatchTST (Nie et al., 2023) and FEDformer (Zhou et al., 2022); 2) CNN-based models: TimesNet (Wu et al., 2023) and MICN (Wang et al., 2022); 3) Linear-based models: TiDE (Das et al., 2023) and DLinear (Zeng et al., 2023). Considering varying look-back window size leads to different performances, we pick up their best performance as baselines, and corresponding results are reported from the original papers.

**Setups** Following Zhou et al. (2021), we normalize the train/val/test sets to zero-mean using the mean and standard deviation from the training set. The Mean Square Error (MSE) and Mean Absolute Error (MAE) are selected as evaluation metrics, consistent with previous methods. All of the models adopt the same prediction length $T = \{96, 192, 336, 720\}$. For the look-back window $t$, we conduct experiments on PDF using $t = 336$ and $t = 720$ while TiDE, PatchTST, and DLinear employ $t = 720, 512, 336$, and all other models use $t = 96$.

### 4.1 MAIN RESULTS

We present multivariate long-term forecasting results in Table 1. Regarding the *Count* value, PDF(720) and PDF(336) achieve the best and second-best results, outperforming all other methods across different categories. Quantitatively, compared with Transformer-based models, PDF(720) yields an overall 14.59% reduction in MSE and 10.77% reduction in MAE. Compared with CNN-based models, PDF(720) yields an overall 24.61% reduction in MSE and 19.91% reduction in MAE.

Table 1: Multivariate long-term forecasting results with different prediction lengths $T \in \{96, 192, 336, 720\}$. The numbers in parentheses next to the method represent the look-back window $t$. The best and the second best results are in **bold** and underlined. The last row *Count* indicates the number of times each method achieves the best or the second best results.

| Catagories | | Ours | | | | Transformers | | | | CNNs | | | | Linears | | | |
|---|---|---|---|---|---|---|---|---|---|---|---|---|---|---|---|---|---|
| Models | | PDF(720) | | PDF(336) | | PatchTST(512) | | FEDformer(96) | | TimesNet(96) | | MICN(96) | | TiDE(720) | | DLinear(336) | |
| Metric | | MSE | MAE | MSE | MAE | MSE | MAE | MSE | MAE | MSE | MAE | MSE | MAE | MSE | MAE | MSE | MAE |
| ETTh1 | 96 | **0.356** | 0.391 | 0.357 | **0.388** | 0.370 | 0.400 | 0.376 | 0.419 | 0.384 | 0.402 | 0.421 | 0.431 | 0.375 | 0.398 | 0.375 | 0.399 |
| | 192 | **0.390** | 0.413 | 0.397 | **0.412** | 0.413 | 0.429 | 0.420 | 0.448 | 0.436 | 0.429 | 0.474 | 0.487 | 0.412 | 0.422 | 0.405 | 0.416 |
| | 336 | **0.402** | **0.421** | 0.409 | 0.422 | 0.422 | 0.440 | 0.459 | 0.465 | 0.491 | 0.469 | 0.569 | 0.551 | 0.435 | 0.433 | 0.439 | 0.443 |
| | 720 | 0.462 | 0.477 | **0.432** | **0.455** | 0.447 | 0.468 | 0.506 | 0.507 | 0.521 | 0.500 | 0.770 | 0.672 | 0.454 | 0.465 | 0.472 | 0.490 |
| ETTh2 | 96 | **0.270** | **0.332** | 0.272 | 0.333 | 0.274 | 0.337 | 0.358 | 0.397 | 0.340 | 0.374 | 0.299 | 0.364 | **0.270** | 0.336 | 0.289 | 0.353 |
| | 192 | 0.334 | **0.375** | 0.335 | **0.375** | 0.341 | 0.382 | 0.429 | 0.439 | 0.402 | 0.414 | 0.441 | 0.454 | **0.332** | 0.380 | 0.383 | 0.418 |
| | 336 | 0.324 | 0.379 | 0.325 | 0.377 | 0.329 | 0.384 | 0.496 | 0.487 | 0.452 | 0.452 | 0.654 | 0.567 | 0.360 | 0.407 | 0.448 | 0.465 |
| | 720 | 0.378 | 0.422 | **0.375** | **0.417** | 0.379 | 0.422 | 0.463 | 0.474 | 0.462 | 0.468 | 0.956 | 0.716 | 0.419 | 0.451 | 0.605 | 0.551 |
| ETTm1 | 96 | **0.277** | 0.337 | 0.280 | **0.335** | 0.293 | 0.346 | 0.379 | 0.419 | 0.338 | 0.375 | 0.316 | 0.362 | 0.306 | 0.349 | 0.299 | 0.343 |
| | 192 | **0.316** | **0.364** | 0.317 | 0.359 | 0.333 | 0.370 | 0.426 | 0.441 | 0.374 | 0.387 | 0.363 | 0.390 | 0.335 | 0.366 | 0.335 | 0.365 |
| | 336 | **0.346** | **0.381** | 0.354 | 0.382 | 0.369 | 0.392 | 0.445 | 0.459 | 0.410 | 0.411 | 0.408 | 0.426 | 0.364 | 0.384 | 0.369 | 0.386 |
| | 720 | **0.402** | **0.409** | 0.405 | 0.413 | 0.416 | 0.420 | 0.543 | 0.490 | 0.478 | 0.450 | 0.481 | 0.476 | 0.413 | 0.413 | 0.425 | 0.421 |
| ETTm2 | 96 | **0.159** | **0.251** | 0.162 | 0.253 | 0.166 | 0.256 | 0.203 | 0.287 | 0.187 | 0.267 | 0.179 | 0.275 | 0.161 | 0.251 | 0.167 | 0.260 |
| | 192 | 0.217 | 0.292 | 0.219 | 0.291 | 0.223 | 0.296 | 0.269 | 0.328 | 0.249 | 0.309 | 0.307 | 0.376 | 0.215 | **0.289** | 0.224 | 0.303 |
| | 336 | **0.266** | 0.325 | 0.270 | 0.326 | 0.274 | 0.329 | 0.325 | 0.366 | 0.321 | 0.351 | 0.325 | 0.388 | 0.267 | 0.326 | 0.281 | 0.342 |
| | 720 | **0.345** | **0.375** | 0.358 | 0.380 | 0.362 | 0.385 | 0.421 | 0.415 | 0.408 | 0.403 | 0.502 | 0.490 | 0.352 | 0.383 | 0.397 | 0.421 |
| Weather | 96 | **0.143** | **0.193** | 0.147 | 0.194 | 0.149 | 0.198 | 0.217 | 0.296 | 0.172 | 0.220 | 0.161 | 0.229 | 0.166 | 0.222 | 0.176 | 0.237 |
| | 192 | **0.188** | **0.236** | 0.192 | 0.239 | 0.194 | 0.241 | 0.276 | 0.336 | 0.219 | 0.261 | 0.220 | 0.281 | 0.209 | 0.263 | 0.220 | 0.282 |
| | 336 | **0.240** | 0.279 | 0.244 | **0.279** | 0.245 | 0.282 | 0.339 | 0.380 | 0.280 | 0.306 | 0.278 | 0.331 | 0.254 | 0.301 | 0.265 | 0.319 |
| | 720 | **0.308** | **0.328** | 0.318 | 0.330 | 0.314 | 0.334 | 0.403 | 0.428 | 0.365 | 0.359 | 0.311 | 0.356 | 0.313 | 0.340 | 0.323 | 0.362 |
| Electricity | 96 | **0.126** | 0.220 | 0.127 | **0.219** | 0.129 | 0.222 | 0.193 | 0.308 | 0.168 | 0.272 | 0.164 | 0.269 | 0.132 | 0.229 | 0.140 | 0.237 |
| | 192 | **0.145** | 0.238 | **0.145** | **0.237** | 0.147 | 0.240 | 0.201 | 0.315 | 0.184 | 0.289 | 0.177 | 0.285 | 0.147 | 0.243 | 0.153 | 0.249 |
| | 336 | **0.159** | **0.255** | 0.162 | **0.255** | 0.163 | 0.259 | 0.214 | 0.329 | 0.198 | 0.300 | 0.193 | 0.304 | 0.161 | 0.261 | 0.169 | 0.267 |
| | 720 | **0.194** | **0.287** | 0.200 | 0.290 | 0.197 | 0.290 | 0.246 | 0.355 | 0.220 | 0.320 | 0.212 | 0.321 | 0.196 | 0.294 | 0.203 | 0.301 |
| Traffic | 96 | 0.350 | 0.239 | 0.351 | **0.238** | 0.360 | 0.249 | 0.587 | 0.366 | 0.593 | 0.321 | 0.519 | 0.309 | **0.336** | 0.253 | 0.410 | 0.282 |
| | 192 | 0.363 | 0.247 | 0.374 | 0.248 | 0.379 | 0.256 | 0.604 | 0.373 | 0.617 | 0.336 | 0.537 | 0.315 | **0.346** | 0.257 | 0.423 | 0.287 |
| | 336 | 0.376 | 0.258 | 0.386 | **0.253** | 0.392 | 0.264 | 0.621 | 0.383 | 0.629 | 0.336 | 0.534 | 0.313 | **0.355** | 0.260 | 0.436 | 0.296 |
| | 720 | 0.419 | 0.279 | 0.421 | 0.278 | 0.432 | 0.286 | 0.626 | 0.382 | 0.640 | 0.350 | 0.577 | 0.325 | **0.386** | **0.273** | 0.466 | 0.315 |
| *Count* | | **52** | | 44 | | 6 | | 0 | | 0 | | 1 | | 20 | | 0 | |

Compared with Linear-based models, PDF(720) yields an overall 7.05% reduction in MSE and 5.51% reduction in MAE. These results affirm that PDF can effectively utilize a long historical look-back window. Furthermore, PDF(720) consistently outperforms all baselines, except for TiDE which exhibits a lower MSE on the traffic dataset. However, this superior performance of TiDE on the traffic dataset is largely attributed to the prior knowledge of static covariates (Das et al., 2023).

## 4.2 EFFECTIVENESS OF PERIOD PATCHING

**Analysis of patch information.** Recent works (Nie et al., 2023; Lin et al., 2023; Zhang et al., 2023) point out that enhancing the semantic information within patches can lead to improved predictions. To assess the performance of patches emphasizing more semantics information versus long-term information, we conduct the following comparative experiments: 1) PatchTST(336): Following the original PatchTST experimental setup, we set each patch length $p = 16$ and stride $s = 8$, yielding a total of 42 patches; 2) PatchTST(336)*: We set $p = 64$, $s = 14$ and obtain 24 patches. Compared with PatchTST(336), each patch is longer and encompasses more semantics information. 3) PDF(336): We employ single-period patching with a period length $p_1 = 24$ and choose $p = s = 1$. Given that $f_1 = 336/p_1 = 14$, each patch has a length of $p \times f_1 = 14$. This resulted in 24 patches, each rich in long-term information.

The experimental results in Table 2 show that compared with PatchTST(336) and PatchTST(336)*, PDF(336) demonstrates noticeable performance improvements on most datasets. These findings emphasize the importance of long-term information contained within the patches. It is noteworthy that both PatchTST(336)* and PDF(336) have the same number of patches. Even though each patch in PatchTST(336)* is longer, theoretically suggesting potential for better prediction results, its performance does not improve and is even worse than PatchTST(336) in some cases. This further indicates that merely extending the semantics information within a patch is not sufficient for enhancing prediction. The key is to ensure each patch captures more long-term information and our period patching method can effectively address this concern.

**Analysis of efficiency.** To further validate the computational efficiency of our period patching approach, we conduct experiments comparing the Multiply-Accumulate Operations (MACs) (Cao et al., 2022) of our PDF with two other patch-based methods across various look-back windows

Table 2: Results of each patch with various semantic information. PatchTST(336)* denotes the variant of PatchTST with longer patches (e.g. more semantics information). The best results are in **bold**.

| Datasets | | ETTh1 | | | | Electricity | | | | Traffic | | | |
|---|---|---|---|---|---|---|---|---|---|---|---|---|---|
| Method | Metric | 96 | 192 | 336 | 720 | 96 | 192 | 336 | 720 | 96 | 192 | 336 | 720 |
| PatchTST(336) | MSE | 0.370 | 0.413 | 0.422 | 0.447 | 0.129 | **0.147** | **0.163** | **0.197** | 0.360 | 0.379 | 0.392 | 0.432 |
| | MAE | 0.400 | 0.429 | 0.440 | 0.468 | **0.222** | **0.240** | 0.259 | **0.290** | 0.249 | 0.256 | 0.264 | 0.286 |
| PatchTST(336)* | MSE | 0.389 | 0.425 | 0.435 | 0.451 | 0.130 | 0.148 | 0.164 | 0.202 | 0.367 | 0.376 | 0.393 | 0.428 |
| | MAE | 0.411 | 0.429 | 0.440 | 0.468 | 0.223 | 0.241 | **0.257** | 0.291 | 0.253 | 0.260 | 0.263 | 0.283 |
| PDF(336) | MSE | **0.357** | **0.397** | **0.409** | **0.432** | **0.128** | **0.147** | 0.166 | 0.204 | **0.353** | **0.374** | **0.388** | **0.423** |
| | MAE | **0.388** | **0.412** | **0.422** | **0.455** | 0.223 | 0.243 | 0.263 | 0.294 | **0.239** | **0.250** | **0.257** | **0.277** |

$t \in \{336, 512, 720, 960\}$ and prediction length $T \in \{96, 192, 336, 720\}$. The results are summarized in Table 3. Overall, the MACs for PDF reduced by 34.64% compared to PatchTST and 74.38% compared to Crossformer. For a fixed look-back window $t$, the increase in MACs for PDF corresponding to the growth in prediction length $T$ typically resides in the magnitude of millions, whereas for PatchTST and Crossformer, it is in the magnitude of gillions. The same observation is noted when keeping the prediction length constant and increasing the size of the look-back window. In extreme cases, specifically for ETTh1 with $t = 960$ and $T = 720$, PDF demonstrated superior lightweight performance, with reductions in MACs of 54.12% and 99.71% compared to PatchTST and Crossformer, respectively.

Table 3: Comparison of Multiply-Accumulate Operations (MACs) among PDF and two other patch-based Transformer methods (PatchTST (Nie et al., 2023) and Crossformer (Zhang & Yan, 2023)) for different look-back window $t \in \{336, 512, 720, 960\}$ and prediction lengths $T \in \{96, 192, 336, 720\}$. "M" and "G" stand for million and gillion operations. The lowest computational costs are in **bold**.

| Look-back | | 336 | | | 512 | | | 720 | | | 960 | | |
|---|---|---|---|---|---|---|---|---|---|---|---|---|---|
| Models | | PDF | Patch. | Cross. | PDF | Patch. | Cross. | PDF | Patch. | Cross. | PDF | Patch. | Cross. |
| ETTh1 | 96 | **3.97M** | 5.21M | 0.81G | **4.67M** | 7.94M | 1.08G | **5.49M** | 11.17M | 1.37G | **6.49M** | 14.89M | 1.72G |
| | 192 | **4.19M** | 5.66M | 1.13G | **5.01M** | 8.63M | 1.40G | **5.98M** | 12.14M | 1.69G | **7.13M** | 16.18M | 2.04G |
| | 336 | **4.53M** | 6.34M | 1.61G | **5.53M** | 9.66M | 1.88G | **6.70M** | 13.59M | 2.17G | **8.10M** | 18.12M | 2.52G |
| | 720 | **5.44M** | 8.15M | 2.89G | **6.91M** | 12.42M | 3.16G | **8.64M** | 17.46M | 3.45G | **10.68M** | 23.28M | 3.80G |
| | Avg. | **4.53M** | 6.34M | 1.61G | **5.53M** | 9.66M | 1.88G | **6.70M** | 13.59M | 2.17G | **8.10M** | 18.12M | 2.52G |
| Electricity | 96 | 3.41G | 5.53G | **2.17G** | 3.60G | 8.42G | **2.89G** | 3.82G | 11.84G | **3.66G** | **4.08G** | 15.79G | 4.59G |
| | 192 | 3.42G | 5.69G | **3.02G** | **3.61G** | 8.67G | 3.74G | **3.84G** | 12.20G | 4.52G | **4.11G** | 16.26G | 5.45G |
| | 336 | **3.43G** | 5.94G | 4.31G | **3.64G** | 9.05G | 5.03G | **3.88G** | 12.73G | 5.81G | **4.16G** | 16.97G | 6.74G |
| | 720 | **3.48G** | 6.60G | 7.74G | **3.70G** | 10.06G | 8.46G | **3.97G** | 14.15G | 9.24G | **4.27G** | 18.86G | 10.17G |
| | Avg. | **3.44G** | 5.94G | 4.31G | **3.64G** | 9.05G | 5.03G | **3.88G** | 12.73G | 5.81G | **4.16G** | 16.97G | 6.74G |
| Traffic | 96 | 9.15G | 14.84G | **5.81G** | 9.67G | 22.61G | **7.74G** | 10.27G | 31.80G | **9.82G** | **10.97G** | 42.40G | 12.32G |
| | 192 | 9.18G | 15.28G | **8.11G** | **9.71G** | 23.29G | 10.04G | **10.33G** | 32.75G | 12.12G | **11.05G** | 43.67G | 14.62G |
| | 336 | **9.22G** | 15.95G | 11.56G | **9.77G** | 24.31G | 13.50G | **10.42G** | 34.18G | 15.57G | **11.17G** | 45.57G | 18.08G |
| | 720 | **9.33G** | 17.73G | 20.77G | **9.94G** | 27.02G | 22.70G | **10.66G** | 37.99G | 24.78G | **11.49G** | 50.66G | 27.28G |
| | Avg. | **9.22G** | 15.95G | 11.56G | **9.77G** | 24.31G | 13.50G | **10.42G** | 34.18G | 15.57G | **11.17G** | 45.58G | 18.08G |

## 4.3 ABLATION STUDIES

**Convolution Module.** To investigate the impact of convolution in short-term variations modeling, we conduct a study comparing the following three cases: 1) Parallel Convolution; 2) Sequential Convolution; 3) Without Convolution. We perform these comparisons in four datasets. The results in Table 4 show that parallel convolution consistently outperforms its sequential counterpart, an advantage possibly stemming from the training challenges posed by deeper networks in serial architectures. Interestingly, models without convolution yield better results than those using sequential convolution, highlighting the drawbacks of overly deep serial networks. Furthermore, when compared to the model without convolution, the parallel approach achieves notable performance improvements on datasets with weaker periodicity, demonstrating its effectiveness in preserving short-term information without increasing network depth. The observed degradation in performance for datasets with strong periodicity, such as Traffic, underscores the necessity of placing emphasis on the long-term variations across periods.

**Variations Aggregation Method.** We explore two methods for aggregating the outputs of multiple DVMBs within the variations aggregation block: 1) Concat: Concatenate the outputs of all DVMBs

Table 4: Ablation study of convolution module in PDF. "Par Conv", "Seq Conv", and "w/o Conv" denote parallel convolution, sequential convolution, and without convolution. The best results are in **bold**.

| Datasets | | ETTh2 | | | | Weather | | | | Electricity | | | | Traffic | | | |
|---|---|---|---|---|---|---|---|---|---|---|---|---|---|---|---|---|---|
| Method | Metric | 96 | 192 | 336 | 720 | 96 | 192 | 336 | 720 | 96 | 192 | 336 | 720 | 96 | 192 | 336 | 720 |
| Par Conv | MSE | **0.270** | **0.334** | **0.324** | **0.378** | **0.143** | **0.188** | **0.240** | 0.308 | **0.126** | 0.145 | **0.159** | **0.194** | 0.360 | **0.363** | 0.376 | **0.419** |
| | MAE | **0.332** | **0.375** | **0.379** | **0.422** | **0.193** | **0.236** | **0.279** | **0.328** | **0.220** | **0.238** | **0.255** | **0.287** | 0.239 | **0.247** | 0.258 | **0.279** |
| Seq Conv | MSE | 0.279 | 0.342 | 0.336 | 0.418 | 0.146 | 0.192 | 0.244 | 0.316 | 0.128 | 0.145 | 0.161 | 0.196 | 0.366 | 0.376 | 0.386 | 0.426 |
| | MAE | 0.339 | 0.381 | 0.391 | 0.452 | 0.197 | 0.241 | 0.283 | 0.336 | 0.225 | 0.241 | 0.257 | 0.289 | 0.255 | 0.260 | 0.267 | 0.287 |
| w/o Conv | MSE | 0.273 | 0.340 | 0.334 | 0.399 | 0.145 | 0.190 | 0.243 | **0.307** | 0.127 | **0.144** | 0.160 | 0.196 | **0.348** | **0.363** | **0.375** | 0.420 |
| | MAE | 0.338 | 0.382 | 0.390 | 0.441 | 0.196 | 0.239 | 0.281 | 0.331 | 0.221 | 0.239 | **0.255** | 0.288 | **0.237** | **0.247** | **0.256** | 0.282 |

and map them through linear projection; 2) Mean: Compute the average outputs of all DVMBs. The experimental results of these two aggregation strategies are presented in Table 5, which shows that the Concat operation generally has better performance than the Mean operation.

Table 5: Ablation study of the variations aggregation method. The best results are in **bold**.

| Datasets | | ETTh2 | | | | ETTm2 | | | | Weather | | | | Electricity | | | |
|---|---|---|---|---|---|---|---|---|---|---|---|---|---|---|---|---|---|
| Method | Metric | 96 | 192 | 336 | 720 | 96 | 192 | 336 | 720 | 96 | 192 | 336 | 720 | 96 | 192 | 336 | 720 |
| Concat | MSE | **0.270** | **0.334** | **0.324** | **0.378** | **0.159** | **0.217** | **0.266** | 0.345 | **0.143** | **0.188** | **0.240** | 0.308 | **0.126** | 0.145 | **0.159** | 0.194 |
| | MAE | **0.332** | **0.375** | **0.379** | **0.422** | **0.251** | **0.292** | **0.325** | **0.375** | **0.193** | **0.236** | **0.279** | **0.328** | **0.220** | **0.238** | **0.255** | **0.287** |
| Mean | MSE | 0.274 | 0.340 | 0.328 | 0.396 | 0.163 | 0.219 | 0.270 | **0.344** | 0.144 | 0.191 | 0.244 | **0.308** | 0.127 | **0.144** | 0.161 | **0.194** |
| | MAE | 0.337 | 0.381 | 0.384 | 0.437 | 0.254 | 0.295 | 0.329 | 0.377 | 0.194 | 0.240 | 0.282 | 0.330 | 0.221 | 0.240 | 0.257 | **0.287** |

## 4.4 COMPUTATIONAL COMPLEXITY ANALYSIS

Table 6 compares the theoretical complexity per layer across different Transformer-based models. The complexity of the encoder layer in the original Transformer is $O(t^2)$. Subsequent works manage to reduce the complexity of the encoder layer to $O(t \log t)$ or even $O(t)$. While the patch-based approaches retain quadratic complexity, the introduction of the patch length $p$ makes $O((\frac{t}{p})^2)$ favorable over $O(t)$ when $t$ is not excessively large. Notably, expect for PDF, all existing Transformer-based methods have the complexities of an encoder layer tied to the length of the look-back window $t$. The computational complexity of PDF is only related to the maximum decoupled periodic length $p_i$. This ensures that even when the $t$ is extremely large, computational costs remain low. For example, if we select the Electricity dataset with $t = 10^5$ and choose its most representative periodic $p_i = 24$ with the patch length $p = 24$, our computational complexity will be significantly lower than all other methods.

Table 6: Theoretical computational complexity per layer in Transformer-based models. $t$ and $T$ denote the length of the look-back window and prediction window, respectively. $d$ denotes the number of variates. $p$ denotes the length of each patch in the patch-based methods.

| Method | Encoder layer | Decoder layer |
|---|---|---|
| Trans. (Vaswani et al., 2017) | $O(t^2)$ | $O(T(t+T))$ |
| In. (Zhou et al., 2021) | $O(t \log t)$ | $O(T(T + \log t))$ |
| Auto. (Wu et al., 2021) | $O(t \log t)$ | $O((\frac{t}{2} + T) \log(\frac{t}{2} + T))$ |
| Pyra. (Liu et al., 2021) | $O(t)$ | $O(t(t+T))$ |
| FED. (Zhou et al., 2022) | $O(t)$ | $O(\frac{t}{2} + T)$ |
| ETS. (Woo et al., 2022) | $O(t \log t)$ | $O(T \log T)$ |
| Cross. (Zhang & Yan, 2023) | $O(\frac{d}{p^2} t^2)$ | $O(\frac{d}{p^2} T(t+T))$ |
| MTP. (Zhang et al., 2023) | $O((\frac{t}{p})^2)$ | $O((\frac{t+T}{p})^2)$ |
| PET. (Lin et al., 2023) | $O((\frac{t}{p})^2)$ | - |
| Patch. (Nie et al., 2023) | $O((\frac{t}{p})^2)$ | - |
| PDF (Ours) | $O((\frac{\max(p_i)}{p})^2)$ | - |

## 5 CONCLUSIONS

This paper introduces an efficient Periodicity Decoupling Framework (PDF) for long-term series forecasting. The PDF captures both short- and long-term temporal variations in 2D spaces. The approach involves breaking down complex 1D time series using a multi-periodic decoupling block (MDB) based on periodicity. Additionally, a dual variations modeling block (DVMB) is proposed to learn short- and long-term variations from the decoupled 2D series in parallel. Compared to previous methods that only model 1D temporal variations, our PDF performs better by effectively extracting both short- and long-term variations. Experiments on real-world datasets demonstrate the superior forecasting performance and computational efficiency over other state-of-the-art methods.

## ACKNOWLEDGMENTS

This work is supported in part by the National Key Research and Development Program of China, under Grant 2022YFF1202104, National Natural Science Foundation of China, under Grant (62302309, 62171248), Shenzhen Science and Technology Program (Grant No. JCYJ20220818101014030, JCYJ20220818101012025, WDZC20231128114058001), Open Fund of National Engineering Laboratory for Big Data System Computing Technology (Grant No. SZU-BDSC-OF2024-23), and Swift Fund Fintech Funding 2023.

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
