# OpenReview forum: "Periodicity Decoupling Framework for Long-term Series Forecasting"
_ICLR.cc/2024/Conference — ICLR 2024 poster_

### Official Review · Reviewer_KqCU · 2023-10-29

**Soundness:** 3 good
**Presentation:** 4 excellent
**Contribution:** 4 excellent
**Rating:** 8
**Confidence:** 4

**Summary:**

This paper proposes a Periodicity Decoupling Framework (PDF) to address the challenges of long-term time series forecasting, which traditionally have intricated temporal patterns. The proposed PDF decouples the time series into distinct short-term and long-term series based on its periodicity. Following this, the dual variations modeling block (DVMB) is employed to extract both short-term and long-term variations. Finally, the variations aggregation block (VAB) aggregates the extracted variations for final predictions. Experimental results show that PDF achieves state-of-the-art performance while maintaining low computational cost.

**Strengths:**

1) Decoupling long-term and short-term variations from complex time series based on periodicity seems reasonable. Besides. the proposed method captures long-term and short-term variations, which fully utilizes the ability of Transformer to model global variations and CNNs to model local variations.
2) The paper proposes a simple yet effective way to extract the dual variations of short-term and long-term variations. Besides, the foundational design is well-motivated and robustly substantiated.
3) The significance of both short-term and long-term variations in time series forecasting is adeptly highlighted. The manuscript is well-written and understandable, and the figures and formulas are well-presented.

**Weaknesses:**

1)Unlike the frequency selection strategy in TimesNet [1], the reason for using a different strategy in the Multi-periodic Decoupling Block needs to be further explained.
2)Were the experimental data in the article averaged over multiple runs with different random seeds? This can enhance the stability of the experimental results.
3)The experimental section mentions that the significant performance gain of TiDE [2] in traffic largely stems from static covariates. It would be preferable to provide experiments to substantiate this claim.

[1] Wu, Haixu, et al. “Timesnet: Temporal 2d-variation modeling for general time series analysis.”, 2022,  https://arxiv.org/pdf/2210.02186.
[2] Das, Abhimanyu, et al. “Long-term Forecasting with TiDE: Time-series Dense Encoder.”, 2023, https://arxiv.org/pdf/2304.08424.

**Questions:**

1) What is the effects of different perodics?  Since the proposed method lies in periodic to decouple the original time series.
2) Can the proposed method applied to other time series applications, like time series classification?

---

> ### Author Response · Authors · 2023-11-16
>
> Thanks for your hard work and valuable comments of this work. Here are responses for your concern and questions:
>
> \
> **Question1:** Unlike the frequency selection strategy in TimesNet, the reason for using a different strategy in the Multi-periodic Decoupling Block needs to be further explained.
>
> **Response1:** Thanks for your suggestions. TimesNet strictly chooses $\text{top-}k$ amplitude values and obtain the most $k$ significant frequencies, which leads to increasing computational burden. Instead, we select frequencies not only focus on high amplitude but also incorporate those with significant values and amplitude. We assert that frequencies with high amplitude better represent the primary components, while those with larger values facilitate a more discernible distinction between long-term and short-term relationships. In this way, our method enhances the decoupling ability of long-term variations and reduces the computational complexity.
>
> \
> **Question2:** Were the experimental data in the article averaged over multiple runs with different random seeds? This can enhance the stability of the experimental results.
>
> **Response2:** Following the settings in the previous work PatchTST, we selected a random seed of 2021 and conducted multiple experiments to ensure a fair comparison, as shown in Table 1 in the paper. Furthermore, as suggested, we have tested our method with two additional random seeds (2022 and 2023), where the results are reported in Table A below, from which we can see that our method performs stably with random seeds.
>
> Table A: Results of MSE/MAE for PDF under different random seeds.
> ||Seed|2021|2022|2023|Avg|
> |-|-|-|-|-|-|
> |ETTh1|96|0.356/0.391|0.357/0.392|0.358/0.390|0.357/0.391|
> ||192|0.390/0.413|0.389/0.412|0.395/0.415|0.391/0.413|
> ||336|0.402/0.421|0.414/0.433|0.407/0.426|0.408/0.427|
> ||720|0.462/0.477|0.470/0.481|0.462/0.476|0.465/0.478|
> |ETTh2|96|0.270/0.332|0.270/0.333|0.271/0.332|0.270/0.332|
> ||192|0.334/0.375|0.334/0.374|0.334/0.375|0.334/0.375|
> ||336|0.324/0.379|0.326/0.382|0.324/0.379|0.325/0.380|
> ||720|0.378/0.422|0.378/0.423|0.378/0.422|0.378/0.422|
> |ETTm1|96|0.277/0.337|0.277/0.337|0.280/0.340|0.278/0.338|
> ||192|0.316/0.364|0.316/0.362|0.317/0.362|0.316/0.363|
> ||336|0.346/0.381|0.346/0.381|0.349/0.380|0.347/0.381|
> ||720|0.402/0.409|0.402/0.409|0.407/0.409|0.404/0.409|
> |ETTm2|96|0.159/0.251|0.161/0.253|0.159/0.251|0.160/0.252|
> ||192|0.217/0.292|0.218/0.292|0.217/0.292|0.217/0.292|
> ||336|0.266/0.325|0.268/0.330|0.266/0.326|0.267/0.327|
> ||720|0.345/0.375|0.344/0.379|0.344/0.376|0.344/0.377|
> |Weather|96|0.143/0.193|0.143/0.195|0.143/0.193|0.143/0.194|
> ||192|0.188/0.239|0.190/0.240|0.189/0.238|0.189/0.239|
> ||336|0.240/0.279|0.242/0.280|0.241/0.279|0.241/0.279|
> ||720|0.308/0.328|0.306/0.327|0.307/0.328|0.307/0.328|
> |Electricity|96|0.126/0.220|0.127/0.221|0.127/0.220|0.127/0.220|
> ||192|0.145/0.238|0.144/0.239|0.145/0.239|0.145/0.239|
> ||336|0.159/0.255|0.160/0.256|0.159/0.255|0.159/0.255|
> ||720|0.194/0.287|0.195/0.288|0.194/0.288|0.194/0.288|
>
> \
> **Question3:** The experimental section mentions that the significant performance gain of TiDE in traffic largely stems from static covariates. It would be preferable to provide experiments to substantiate this claim.
>
> **Response3:** Thanks for your suggestions. We removed the predefined static covariates in TiDE and reimplemented it on the traffic dataset, where the results are shown in Table B. It can be seen that the performance of TiDE without static covariates still performs worse than our method.
>
> Table B: Results of MSE/MAE with PDF and TiDE (without static covariates) on the Traffic dataset.
> || Model |PDF|TiDE (w/o static covariates)|
> |-|-|-|-|
> |Traffic|96|0.350/0.239|0.368/0.257|
> ||192|0.363/0.247|0.382/0.262|
> ||336|0.376/0.258|0.395/0.268|
> ||720|0.419/0.279|0.434/0.289|
>
> \
> **Question4:** What are the effects of different periods? Since the proposed method lies in periodic to decouple the original time series.
>
> **Response4:** Long time series generally exhibit multi-periodicities. By decoupling different periods into 2D tensors with varying periods, we can theoretically transmute data into multiple resolutions. Table C below provides performance comparison between a single period ($k=1$) and five periods ($k=5$). The results show that the selection of multiple periods effectively enhances the predictive performance of the model.
>
> Table C: Results of MSE/MAE with varied period counts ($k$) on the Electricity dataset.
> |||$k=1$|$k=5$|
> |-|-|-|-|
> |Electricity|96|0.128/0.222|0.126/0.220|
> ||192|0.146/0.242|0.145/0.238|
> ||336|0.162/0.260|0.159/0.255|
> ||720|0.200/0.292|0.194/0.287|
>
> \
> **Question5:** Can the proposed method applied to other time series applications, like time series classification?
>
> **Response5:** Yes! Although our method is designed for modeling long time sequences, it can also be applied to classification tasks by modifying the prediction head. We will explore the potential applications of our method in future works.

---

> > ### Comment · Reviewer_KqCU · 2023-11-21
> >
> > Thanks for your responses. My concerns have been solved and I have no further questions.

---

### Official Review · Reviewer_Puj1 · 2023-10-31

**Soundness:** 2 fair
**Presentation:** 2 fair
**Contribution:** 2 fair
**Rating:** 3
**Confidence:** 4

**Summary:**

This paper introduces a novel approach called the Periodicity Decoupling Framework (PDF) for enhancing time series forecasting. The PDF method comprises three main components: the multi-periodic decoupling block (MDB) to extract 2D temporal variations from 1D time series, the dual variations modeling block (DVMB) for capturing short-term and long-term variations, and the variations aggregation block (VAB) for making predictions. Extensive experiments on seven real-world long-term time series datasets demonstrate the effectiveness of proposed methods.

**Strengths:**

1. this paper is easy to follow.
2. this paper studies a classic problem, time series forecasting.

**Weaknesses:**

1. the motivation. This paper is not well-motivated. Why do we need 2D states to capture periodicity in time series forecasting? Do the authors show the necessity of the 2D modeling for periods?
2. the novelty is limited. The biggest contribution of this paper, the formulation of 1D to 2D transformation seems to be the same as TimesNet [1], which makes the novelty largely limited.
3. the contribution is a little weak. Compared with existing works (such as TimesNet), it only revises several blocks for time series forecasting. The methods don't contribute well to the community.
4. missing of related work. More periodic modeling works should be discussed [2,3]. Also, differences compared with TimesNet should be discussed.

[1] Timesnet: Temporal 2d-variation modeling for general time series analysis. In ICLR.
[2] DEPTS: Deep Expansion Learning for Periodic Time Series Forecasting. In ICLR.
[3] Bridging self-attention and time series decomposition for periodic forecasting. In CIKM.

**Questions:**

See weakness.

---

> ### Author Response · Authors · 2023-11-16
>
> Thanks for your hard work and valuable comments of this work. Here are responses for your concern and questions:
>
> \
> **Question1:** The motivation. This paper is not well-motivated. Why do we need 2D states to capture periodicity in time series forecasting? Do the authors show the necessity of the 2D modeling for periods?
>
> **Response1:** Thanks for your comments. The necessity of the 2D modeling lies in the following reasons: 1) 2D tensors have more compact representations than 1D time series, which has been shown in TimesNet; 2) The decouple difficulty of time series lies in how to capture long-term variations. In fact, transforming the complex 1D series into simpler 2D tensors with various periodicity provides a proper solution. We would add more discussions to make our motivation more clear.
>
> \
> **Question2:** The novelty is limited. The biggest contribution of this paper, the formulation of 1D to 2D transformation seems to be the same as TimesNet, which makes the novelty largely limited.
>
> **Response2:** Thanks for your comments. In fact, **our core contribution lies in the proposed novel Periodicity Decoupling Framework (PDF) to decouple and model temporal variations**, rather than the formulation 1D to 2D transformation. We are encouraged by the positive comments about novelty (Reviewer fZ8f) and interesting work (Reviewer gS6P), well-motivated and reasonable design (Reviewers gS6P, KqCU), and sufficient experiments (Reviewers fZ8f, gS6P) of our work.
>
> As pointed out, we borrow the idea of utilizing periodicity in TimesNet to develop our Multi-periodic Decoupling Block to decouple into simpler short- and long-term series. However, there exist several major differences between our method and TimesNet as below:
> 1. The way of modeling variations is different. TimesNet heavily relies on convolutional neural networks (CNN) to extract temporal variations, which limits the receptive field size of TimesNet due to the use of CNNs. Instead, we introduced novel slicing and patching operations to the 2D tensors, contributing to subsequent CNN and Transformer to extract the decoupled short-term and long-term variations.
> 2. The frequency selection strategy of capturing periodicity is different. TimesNet strictly chooses top-k amplitude values and obtain the most k significant frequencies. Instead, we select frequencies not only focus on high amplitude but also incorporate those with significant values and amplitude. We assert that frequencies with high amplitude better represent the primary components, while those with larger values facilitate a more discernible distinction between long-term and short-term relationships (More details are in Section 3.2).
> 3. The time complexity varies much. For longer inputs, TimesNet will significantly increase computational cost, as it requires stacking multiple convolutional kernels to achieve an adaptable receptive field. In contrast, our method employs a computation-friendly patch-based approach for modeling long-term information, where the computational complexity is related only to the period length (see Table 6).
>
> Lastly, we believe that our experimental results can show the effectiveness of our method. Table 1 highlights the success in long-term series forecasting, while Tables 2 and 4 show the efficacy of our decoupling approach and architectural design. Figure 1(b) and Table 3 illustrate the low complexity of our method. We would clarify the differences between our method and TimesNet in the revision.
>
> \
> **Question3:** The contribution is a little weak. Compared with existing works (such as TimesNet), it only revises several blocks for time series forecasting. The methods don't contribute well to the community.
>
> **Response3:** Thanks for your comments. In fact, our method is not merely a modification of several blocks in time series forecasting, yet our core contribution lies in the proposed novel Periodicity Decoupling Framework to decouple and model temporal variations. Compared with TimesNet, there exist several differences: 1) The novel slicing and patching operations, and modeling variations are different;  2) The frequency selection strategy of capturing periodicity is different;  3) The time complexity varies much.
>
> \
> **Question4:** Missing of related work. More periodic modeling works should be discussed. Also, differences compared with TimesNet should be discussed.
>
> **Response4:** Thank you for pointing it out. The previous two works [2][3] focus on addressing the inherent periodic properties, which may be helpful for capturing periodicity accurately for our method. We will further explore the potential applications of the inherent periodic properties in future works.
>
> Compared with TimesNet, there exist several differences: 1) The novel slicing and patching operations, and modeling variations are different; 2) The frequency selection strategy of capturing periodicity is different;  3) The time complexity varies much. We would add such discussions in the revision.

---

> > ### Author Response · Authors · 2023-11-23
> > **Reminder for post-rebuttal feedback**
> >
> > Dear Reviewer Puj1,
> >
> >  We greatly appreciate your initial valuable comments. We hope that you could have a quick look at our responses to your concerns. It would be highly appreciated if you could kindly update the initial rating if your questions have been addressed. We are also happy to answer any additional questions before the rebuttal ends.
> >
> > Best Regards,
> > Paper2303 Authors

---

> ### Author Response · Authors · 2023-11-22
> **Rebuttal for feedback**
>
> We would like to thank the reviewer for the helpful discussion during the first round of the review. We hope our response has adequately addressed your concerns. We take this as a great opportunity to improve our work and shall be grateful for any additional feedback you could give to us.

---

### Official Review · Reviewer_gS6P · 2023-10-31

**Soundness:** 3 good
**Presentation:** 4 excellent
**Contribution:** 4 excellent
**Rating:** 8
**Confidence:** 4

**Summary:**

Real-world time series forecasting is challenging, since it usually contains intricate temporal patterns. This paper focuses on exploits intricate temporal patterns by decoupling the complex series into simpler series to achieve long-term series forecasting. Thus, the authors developed  a novel Periodicity Decoupling Framework (PDF) for long-term series forecasting by capturing 2D temporal variation modeling. Extensive experimental results across  seven real-world long-term time series datasets demonstrate the superiority of the proposed method over other state-of-the-art methods, in terms of both forecasting performance and computational efficiency.

**Strengths:**

- It is reasonable and interesting to decouple the complex 1D time series into simpler series with various variations based on periodicity.
- The proposed multi-periodic decoupling block is a novel and effective solution to capture various periods of the input series. Based on the periodicity of the time series, the 1D time series are decoupled into simpler short- and long-term series.
- Extensive experiments demonstrate the effectiveness of the proposed over other state-of-the-art methods (e.g., TimesNet, TiDE) across various long-term time series datasets.
- The overall paper is well-written and easy to follow.

**Weaknesses:**

- How to decouple the time series remains an open question. Although the authors propose a simple yet effective periodicity-based strategy to decouple the time series, how to evaluate the effectiveness of decouple strategy has been less explained.
- Experiments show the computational efficiency of the proposed methods over other transformer-based methods. However, the authors have not compared the running time of different methods.

**Questions:**

- Is there other ways to decouple time series? Besides, how to evaluate the effectiveness of the proposed decoupled strategy?
- It is recommended to compare the running time of other methods.

---

> ### Author Response · Authors · 2023-11-16
>
> Thanks for your hard work and valuable comments of this work. Here are responses for your concerns and questions:
>
> \
> **Question1:** How to decouple the time series remains an open question. Although the authors propose a simple yet effective periodicity-based strategy to decouple the time series, how to evaluate the effectiveness of decouple strategy has been less explained.
>
> **Response1:** Thank you for your valuable suggestions. In theory, there are various ways to evaluate the effectiveness of decoupling strategies. The most straightforward is to compare the performance (e.g. MSE and MAE) of models constructed using various decoupling strategies or assessing their computational complexity， which we adopts for evaluating our method (More details can be found in Appendix D).  Another  feasible approach may be to separately compute the similarity between the decoupled long-term or short-term variations and the original sequence or subsequence.
>
> \
> **Question2:** Experiments show the computational efficiency of the proposed methods over other transformer-based methods. However, the authors have not compared the running time of different methods.
>
> **Response2:** Thanks for your suggestions. In fact, we have reported the Multiply-Accumulate Operations (MACs) among PDF and two other patch-based Transformer methods (PatchTST[1] and Crossformer[2]) in Table 3 in the paper.  As suggested, we have compared the MACs results of two additional CNN-based methods (TimesNet[3] and MICN[4])  In Table A below.
>
> Table A: Results of Multiply-Accumulate Operations (MACs) for TimesNet and MICN.
> |   |   Model             | TimesNet(960) | MICN(960) | TimesNet(720) | MICN(720) | TimesNet(512) | MICN(512) | TimesNet(336) | MICN(336) |
> |-------------|-----------|---------------|-----------|---------------|-----------|---------------|-----------|---------------|-----------|
> | ETTh1       | 96        | 3.12 G        | 14.46 G   | 2.40 G        | 10.64 G   | 1.83 G        | 7.56 G    | 1.27 G        | 5.22 G    |
> |             | 192       | 3.40 G        | 16.08 G   | 2.69 G        | 12.13 G   | 2.07 G        | 8.93 G    | 1.61 G        | 6.50 G    |
> |             | 336       | 3.84 G        | 18.61 G   | 3.12 G        | 14.46 G   | 2.49 G        | 11.08 G   | 1.98 G        | 8.51 G    |
> |             | 720       | 4.96 G        | 25.93 G   | 4.28 G        | 21.26 G   | 3.64 G        | 17.41 G   | 3.10 G        | 14.46 G   |
> | Electricity | 96        | 792.35 G      | 15.17 G   | 626.61 G      | 11.19 G   | 466.17 G      | 7.96 G    | 325.94 G      | 5.50 G    |
> |             | 192       | 864.74 G      | 16.88 G   | 686.32 G      | 12.76 G   | 529.02 G      | 9.41 G    | 402.06 G      | 6.86 G    |
> |             | 336       | 972.98 G      | 19.55 G   | 792.38 G      | 15.22 G   | 636.42 G      | 11.68 G   | 508.55 G      | 8.98 G    |
> |             | 720       | 1260.42 G     | 27.23 G   | 1080.25 G     | 22.35 G   | 925.77 G      | 18.32 G   | 792.63 G      | 15.22 G   |
> | Traffic     | 96        | 1586.96 G     | 16.39 G   | 1254.32 G     | 12.13 G   | 951.06 G      | 8.66 G    | 648.51 G      | 6.00 G    |
> |             | 192       | 1732.35 G     | 18.26 G   | 1399.70 G     | 13.85 G   | 1080.39 G     | 10.24 G   | 793.72 G      | 7.48 G    |
> |             | 336       | 1947.07 G     | 21.16 G   | 1587.26 G     | 16.52 G   | 1284.12 G     | 12.72 G   | 1009.44 G     | 9.78 G    |
> |             | 720       | 2522.05 G     | 29.46 G   | 2161.59 G     | 24.22 G   | 1853.17 G     | 19.88 G   | 1649.38 G     | 16.52 G   |
>
> \
> [1] Nie, Yuqi, et al. "A time series is worth 64 words: Long-term forecasting with transformers.", 2022, https://arxiv.org/pdf/2211.14730.
>
> [2] Zhang, Yunhao, and Junchi Yan. "Crossformer: Transformer utilizing cross-dimension dependency for multivariate time series forecasting.", 2022, https://openreview.net/pdf?id=vSVLM2j9eie.
>
> [3] Wu, Haixu, et al. "TimesNet: Temporal 2D-Variation Modeling for General Time Series Analysis.", 2022, https://arxiv.org/abs/2210.02186.
>
> [4] Wang, Huiqiang, et al. "MICN: Multi-scale local and global context modeling for long-term series forecasting.", 2022, https://openreview.net/pdf?id=zt53IDUR1U.

---

> > ### Comment · Reviewer_gS6P · 2023-11-23
> >
> > Thank you for your replies. I have no further questions and my concerns have been addressed.

---

### Official Review · Reviewer_fZ8f · 2023-10-31

**Soundness:** 3 good
**Presentation:** 3 good
**Contribution:** 4 excellent
**Rating:** 8
**Confidence:** 4

**Summary:**

This paper decouples variations of different scales in multi-variate long-term time series based on their periodicity. It then leverages the respective modeling strengths of CNNs and Transformer models to represent these scale-distinct variations. Extensive experiments on multiple long-term time series forecasting datasets demonstrate that the proposed Periodic Decoupling Framework (PDF) method outperforms the latest state-of-the-art approaches across various forecasting time intervals.

**Strengths:**

Strengths:
1) The proposed periodicity decoupling framework is a novel and efficient solution to capture 2D temporal variation modeling for long-term series forecasting. Besides, modeling long-term and short-term variations also offers a new perspective for time series forecasting.
2) The fusion of CNNs and Transformers is prevalent in CV and NLP. This paper bridges time series studies with other advanced domains, introducing the PDF framework by leveraging the strengths of both architectures. The PDF demonstrates outstanding performance and efficiency in time series tasks, as evidenced by thorough experiments.
3) The authors analyze the computational burden with other state-of-the-art methods and demonstrate the efficiency of the proposed method.
4) The overall paper is well-organized and easy to follow.

**Weaknesses:**

1) The channel-independent strategy focuses on the dependencies of the time dimension adopted in this work and is inspired by PatchTST [1]. However, channel dependency is also helpful. In other works, such as Crossformer [2], modeling channels have also been beneficial to some extent in forecasting results. A more robust justification for this choice is needed.
2) The innovativeness of the patching method in this work needs to be articulated more specifically. What distinguishes this patching approach from other recent patch-based works (such as PatchTST [1] and PETformer [3])?

Other minor suggestions：
In the first sentence of Section 4.4, one of the occurrences of "complexity" should be removed.

[1] Nie, Yuqi, et al. "A time series is worth 64 words: Long-term forecasting with transformers.", 2022, https://arxiv.org/pdf/2211.14730.
[2] Zhang, Yunhao, and Junchi Yan. "Crossformer: Transformer utilizing cross-dimension dependency for multivariate time series forecasting.", 2022, https://openreview.net/pdf?id=vSVLM2j9eie.
[3] Lin, Shengsheng, et al. "PETformer: Long-term time series forecasting via placeholder-enhanced transformer.", 2023, https://arxiv.org/pdf/2308.04791.

**Questions:**

1) What are the benefits of long-term variation and short-term variation extractors?
2) The periodicity obtained is adaptive to the time series of input?
3) It is reasonable to model 2D temporal short-term and long-term variations. Is it possible to model 3D variations in time series?

---

> ### Author Response · Authors · 2023-11-16
>
> Thanks for your hard work and valuable comments of this work. Here are responses for your concerns and questions:
>
> \
> **Question 1:** The channel-independent strategy focuses on the dependencies of the time dimension adopted in this work and is inspired by PatchTST [1]. However, channel dependency is also helpful. In other works, such as Crossformer [2], modeling channels have also been beneficial to some extent in forecasting results. A more robust justification for this choice is needed.
>
> **Response 1:** Thanks for your valuable comments and suggestions. As is shown in the previous work [4], the channel-independent strategy can better mitigate data drift. While channel dependencies theoretically have the potential to bring about information gain, in practice, data shifts between channels can pose also challenging modeling difficulties.
>
> \
> **Questions2:** The innovativeness of the patching method in this work needs to be articulated more specifically. What distinguishes this patching approach from other recent patch-based works (such as PatchTST [1] and PETformer [3])?
>
> **Response2:** Our patching operation is designed to efficiently capture long-term dependencies in time series. We perform patching operations on identical time segments across all periods, while other approaches such as PatchTST and PETformer apply patching to adjacent time stamps. In comparison, our approach can effectively preserve more long-term information in each patch while reducing computational complexity, as shown in Table 2 and Table 6 in the paper.
>
> \
> **Question3:** What are the benefits of long-term variation and short-term variation extractors?
>
> **Response3:** Long-term and short-term variation extractors are employed to independently model dependencies at different scales. In this way, it is beneficial for extractors to not only take into account the long-term and short-term characteristics of the input data but also to reduce the computational costs.
>
> \
> **Question4:** The periodicity obtained is adaptive to the time series of input?
>
> **Response4:** The obtained periodicity is not adaptive for our method, since the adaptive nature of periodic scales may result in inconsistent 2D shape dimensions in the input. We will explore the adaptive nature of periodic scales in future works.
>
> \
> **Question5:** It is reasonable to model 2D temporal short-term and long-term variations. Is it possible to model 3D variations in time series?
>
> **Response5:** Thanks for your suggestions. There may exist several ways of modeling 3D variations. 1) we can consider dependencies between channels, as is explored for our method; 2) we can further decompose temporal variations to model 3D variations. For example, we may use wavelet transformations to capture the dynamic periodicity of the data, which is worth exploring in future works.
>
> \
> [1] Nie, Yuqi, et al. "A time series is worth 64 words: Long-term forecasting with transformers.", 2022, https://arxiv.org/pdf/2211.14730.
>
> [2] Zhang, Yunhao, and Junchi Yan. "Crossformer: Transformer utilizing cross-dimension dependency for multivariate time series forecasting.", 2022, https://openreview.net/pdf?id=vSVLM2j9eie.
>
> [3] Lin, Shengsheng, et al. "PETformer: Long-term time series forecasting via placeholder-enhanced transformer.", 2023, https://arxiv.org/pdf/2308.04791.
>
> [4] Lu Han, et al. "The Capacity and Robustness Trade-off: Revisiting the Channel Independent Strategy for Multivariate Time Series Forecasting", 2023, https://arxiv.org/abs/2304.05206

---

> > ### Comment · Reviewer_fZ8f · 2023-11-23
> >
> > I appreciate your thoughtful and detailed replies to my questions and comments. You have addressed all of my concerns satisfactorily and I have no further queries.

---

### Comment · Area_Chair_wxKf · 2023-11-23
**[ICLR 2024 Reviewers’ feedback] Please read authors’ responses and give your feedback**

Dear Reviewers,

Thanks again for your strong support and contribution as an ICLR 2024 reviewer.

Please check the response and other reviewers’ comments. You are encouraged to give authors your feedback after reading their responses. Thanks again for your help!

Best,

AC

---

### Meta-Review · Area_Chair_wxKf · 2023-12-12

**Metareview:**

This paper proposes a novel and efficient periodicity decoupling framework to capture 2D temporal variation modeling for long-term series forecasting.  Extensive experiments demonstrate the effectiveness of the proposed method over other state-of-the-art methods across various long-term time series datasets. Most reviewers give high scores. Only one reviewer gave a negative score and did not reply to the authors' rebuttal. The responses have addressed most of their concerns.

**Justification For Why Not Higher Score:**

One reviewer has some concerns about motivation and contributions.

**Justification For Why Not Lower Score:**

Most reviewers give high ratings.

---

### Decision · Program_Chairs · 2024-01-16

Accept (poster)